# Experimental Study of Yield Surface in Polypropylene Considering Rate Effect: Stress Path Dependence

**DOI:** 10.3390/polym14153225

**Published:** 2022-08-08

**Authors:** Hang Zheng, Xiaofei Yi, Zhiping Tang, Kai Zhao, Yongliang Zhang

**Affiliations:** CAS Key Laboratory of Mechanical Behavior and Design of Materials, Department of Modern Mechanics, University of Science and Technology of China, Hefei 230027, China

**Keywords:** polypropylene, shear-compression, strain-rate effect, stress path, yield surface

## Abstract

In order to investigate the yield surface evolution of polypropylene (PP) under dynamic impact and the relationship between yield surface parameters and the strain rate, five shear-compression specimens (SCSs) with different inclination angles are designed and produced to explore the yield behavior of PP under dynamic loading. Dynamic combined stress loading paths with different compression-shear ratios are achieved by the split Hopkinson pressure bar (SHPB). The evolution laws of the compressive stress and shear stress in the measurement region during the PP SCS compressive deformation process are analyzed. In terms of mechanical response, PP under combined compression-shear loading is of visco-elasticity plasticity and its deformation undergoes a three-stage transition, namely “unyield→yield→failure”. The yield characteristics of PP are found to be affected not only by the hydrostatic pressure but also by the stress path. According to the Hu–Pae yield criterion, the dynamic yield surface and model parameters of PP are obtained, and the relationship between the yield surface and the strain rate is ascertained. These findings contribute to deepening the research on the mechanical response characteristics of PP-based materials.

## 1. Introduction

With such excellent properties as lightweight, high strength, and easy processing, PP has enjoyed wide applications in engineering [1]. In actual production, processing, and service, PP failure and damage generally occur in a complex stress state that is different from the deformation state where the material is in uniaxial tension tests. Therefore, it is of great necessity and important academic significance to identify the effect of the strain rate on PP deformation under combined compression-shear loading and to analyze the yield criterion and yield surface evolution characteristics of PP under complex stress conditions.

As is known, the yielding behavior of polymers such as PP, polyethylene, nylon and polymer nanocomposites exhibit evident hydrostatic pressure dependence and strain rate sensitivity under external loading [2,3,4,5,6,7,8,9,10]. Nonetheless, the experiments on different types of polymers have revealed the impotence of both the von Mises and Tresca yield criteria in explaining observed yielding behavior. Eyring [2] investigated the effect of the strain rate and temperature on the yield strength of amorphous polymer materials. Mears et al. [3] experimentally studied the tensile yield law of PE and PP under different hydrostatic pressures, discussed the relationship between yield stress and hydrostatic pressure, and offered parameters for the two criteria involving yield and hydrostatic pressure, namely the Mohr–Coulomb yield criterion and the Hu–Pae yield criterion. Silano et al. [4] experimentally explored the shear deformation and yield behavior of polyoxymethylene (POM) and PP under hydrostatic pressure and concluded that the relationship between shear yield and hydrostatic pressure in POM followed the Hu–Pae rather than the von Mises yield criterion. Duan et al. [7] and Jeridi et al. [8] described the yielding behavior of polymers in wide loading rate and temperature ranges using the Johnson–Cook model and a multiple motion mechanisms-based model, respectively.

In order to study the yield characteristics of polymers under complex stress loading and construct yield surfaces, combined stress should be loaded onto the specimens via different stress loading paths to make them yield. In view of this, Farrokh et al. [11] performed stress loading experiments on Nylon 101 and developed a strain rate-dependent yield criterion, but to the neglect of the effect of loading paths. Zhou et al. [12] designed an inclined loading pad in an axisymmetric shape, which applied quasi-static and dynamic combined compression-shear loading to metal and rock materials. In addition, they experimentally studied the compression-shear yield characteristics of heterogeneous materials using the SHPB and found that the specimen slipped in case of a large angle of inclination, resulting in unreliable shear stress results. Zhou et al. [13] investigated the failure behavior of PMMA in the compression-shear stress state using a complex loading device with a variable-angle pressure head and carried out experiments under different combined compression-shear loading achieved by changing the angle of the pressure head. Such a method that achieves combined compression-shear loading by a specially designed loading device turns out difficult to achieve combined loading path adjustment. What is worse, for the method, special fixtures are required; otherwise, the specimen will slide under a large inclination angle, leading to the failure in determining material yield.

In order to avoid the use of complex loading devices, another way to implement combined compression-shear loading is to employ special experimental geometries. For example, Meyer et al. [14] and Andrade et al. [15] designed hat-shaped specimens to emphatically study the adiabatic shear failure of materials under high-speed shearing. The deformation of the specimens was found to concentrate near the adiabatic shear band, thereby failing to accurately reflect material response and yield characteristics at large strains. The SCS, developed by Rittel et al. [16], features a new specimen geometry and consists of a cylinder in which two diametrically opposed slots are machined at a 45° angle with respect to the longitudinal axis. On the basis of this experimental geometry, Zhao et al. [17] developed a modified SCS to investigate the compression-shear properties of polyurea. Alkhader et al. [18] carried out dynamic compression-shear tests on polyurea using a modified SCS geometry, ascertaining the strain rate sensitivity of the material and identifying the effect of pressure on shear response. With the SCS, Jin et al. [19,20,21] studied the mechanical responses of PMMA and PA66 under different compression-shear combinations and characterized the yield of various samples under quasi-static loading.

In summary, there are limited experimental data on the compression-shear loading paths of polymers represented by PP. Additionally, the applicability of yield criteria and the evolution law of yield surfaces of polymers in a complex stress state remain unclear. Hence, in order to investigate the evolution law of PP’s yield surface under dynamic impact and its relationship with the strain rate, this study designs five SCSs with different inclination angles to look into the yield behavior of PP under dynamic loading and achieves dynamic combined stress loading paths under different ratios of compression-shear stress using the SHPB. Furthermore, the dynamic yield surface of the specimens is obtained according to the hydrostatic pressure-dependent Hu–Pae yield criterion, and their rate effect is analyzed as well.

## 2. Experimental

### 2.1. Specimen

The PP used in this study was manufactured by Hefei Huasheng Rubber Co., Ltd., and its parameters are shown in Table 1. In order to investigate the yielding behavior of PP under combined shear-compression loading, shear-compression loading was achieved using the SCSs, as shown in Figure 1. The SCSs were prepared by machining two symmetrical inclined parallel slots on both sides of the cylindrical specimens, which were standardized as the rigid plastics of ASTM D695 [22]. The five inclination angles of the SCSs were set θ = 5°, 15°, 30°, 45°, 60°, as shown in Table 2, so as to identify the mechanical responses of PP under different loading paths.

### 2.2. Experimental Method

The experiments were carried out on an SHPB with a diameter of 20 mm, whose schematic diagram is shown in Figure 2. When the bullet passes through the laser light path and block the laser light in turn, so that the photoelectric sensor can obtain the signal. The speed *V*_0_ can be measured by using two photoelectric sensors at different positions. After the bullet hits the incident bar, the stress wave propagates to the SCS specimen, transmission bar and absorption bar in turn, and the reflection occurs at the interface. Table 3 presents the parameters of the bar. The key to the dynamic experiments was to attach high signal-to-noise ratio strain gauges to both the incident and transmission bars so that the load force and displacement signals at both ends of the specimens were obtained. The subscripts *i*, *t*, and *r* denote the incident, transmitted, and reflected strain signals, respectively.

According to the incident strain $\varepsilon_{i}(t)$ transmitted strain $\varepsilon_{t}(t)$, and reflected strain $\varepsilon_{r}(t)$, the load force *P*(*t*) and displacement *d*(*t*) at both ends of the specimens were calculated. Specifically, the load force *P*(*t*) is given by:(1)$$\left\{ \begin{array}{c} P_{i}t=EA\left[\varepsilon_{i}(t)+\varepsilon_{r}(t)\right]\\ P_{t}t=EA\varepsilon_{t}(t) \end{array}\right.$$

The displacement *d*(*t*) is given by:(2)$$\left\{ \begin{array}{c} u_{i}(t)=c\int_{0}^{t}\left[\varepsilon_{i}(t)-\varepsilon_{r}(t)\right]dt\\ u_{t}(t)=c\int_{0}^{t}\varepsilon_{t}(t)dt \end{array}\right.$$

Hence,
(3)$$\left\{ \begin{array}{l} d(t)=u_{i}(t)-u_{t}(t)\\ P(t)=\frac{P_{i}(t)+P_{t}(t)}{2} \end{array}\right.$$

According to the equations in System (4), the force–time relationship of the specimen can be decomposed into the normal stress–time relationship and the shear stress–time relationship. Meanwhile, according to the equations in System (5), the displacement–time relationship can be decomposed into the compressive strain–time relationship and the shear strain–time relationship [6].
(4)$$\left\{ \begin{array}{c} \sigma_{yy}=\frac{P}{Rd}\cos^{2}\theta \\ \sigma_{xy}=\frac{P}{Rd}\cos\theta \sin\theta \end{array}\right.$$
(5)$$\left\{ \begin{array}{l} {\dot{\varepsilon }}_{yy}=\frac{\dot{\delta }}{h}\frac{\cos\theta }{4\sin^{2}\theta +\cos^{2}\theta }\\ {\dot{\varepsilon }}_{xy}=2\frac{\dot{\delta }}{h}\frac{\sin\theta }{4\sin^{2}\theta +\cos^{2}\theta } \end{array}\right.$$

### 2.3. Results

A set of experiments where the five SCSs with different inclination angles were subjected to impact loading of different velocities were conducted to investigate the yield surface evolution law of PP under the impact, as well as the relationship between yield surface parameters and the strain rate. Here let us take the 15° SCS as an example to expound on the data processing method in the tests. The impact test for the 15° SCS was carried out a total of seven times, with an impact velocity ranging from 1.88 m/s to 4.47 m/s. The experimental waveform is shown in Figure 3. The data in Figure 3 were processed to obtain the force–time curves (Figure 4) and force–displacement curves (Figure 5) of the 15° SCS under different impact velocities, on the basis of which the normal stress–strain curves (Figure 6) and tangential stress–strain curves (Figure 7) were drawn.

The yield loads at different impact velocities and the displacements corresponding to the loads were obtained according to Figure 5, while the normal and tangential yield stress components were worked out according to Figure 6 and Figure 7, respectively. In the case that the specimen was damaged, as shown by the experimental results when the impact velocities were 4.24 m/s and 4.47 m/s, the failure load and the corresponding displacement, as well as the normal failure stress component and the tangential failure stress component, could be obtained. For the SCSs with other inclination angles, the dynamic experimental results were processed in a similar way to the 15° SCS. The specific parameters and results of all dynamic experiments are listed in Table 4.

As revealed by the force–time curves under different impact velocities in Figure 4, the transmitted signal underwent no significant fluctuation when the impact velocity was 1.88 m/s. The force–displacement curves of the SCS obtained are shown in Figure 5. At the impact velocity of 1.88 m/s, the maximum force of the specimen was 1858 N, less than 2350 N, with the yield force of the 15° SCS in the quasi-static compression test [23]. This suggested that the material did not yield under the impact velocity. With regard to the SCS experimental signals under other impact velocities, the increased impact velocity led to the growing stress transmitted to the output bar and the intensified semiconductor strain gauge signals thereon as a result.

### 2.4. Stress Oscillation

As can be seen from Figure 4, at about 23 us after the impact loading, the stress of the 15° SCS began to oscillate, which should be attributed to the long length of the specimen, which resulted in the back-and-forth propagation of the stress wave in the specimen. Considering the structural characteristics, the SCS should be regarded as a structural member with a changing cross-section during the impact process. Consequently, in the process of stress wave propagation, all the SCSs could be considered to have three regions with different wave impedance, which are named as zone I, zone II, and zone III, respectively. At the region boundaries, the transmission and reflection of stress waves occurred. According to the generalized wave impedance, the transmission and reflection coefficients on the four interfaces were calculated, as shown in Table 5. Figure 8 presents the wave system diagrams of the specimen, the incident bar, and the transmission bar during the loading process when the intensity of the incident wave is 100 MPa. As illustrated, after three cycles of propagation in the SCS II region, the intensity of the stress wave on the transmission bar saw a drop, which was manifest in the experiment as stress oscillation.

Alkhder et al. [15] applied the SCS geometry to the dynamic testing of polyurea and found the presence of stress oscillation. Dorogoy [24] performed finite element analysis of the dynamic loading process of an aluminum SCS using LS-DYNA and discovered that the SCS in the impact loading process and the stress signal in the plateau phase underwent obvious oscillations. In order to analyze the stress oscillation PP SCS experienced during the loading process in the dynamic experiment, the Abauqs-explicit finite element code was utilized to numerically simulate the impact process of the SCS made of an elastoplastic material, and the finite element model parameters are listed in Table 6. The stress signal of the transmission bar was obtained by the simulation, as shown in Figure 8. The results show that even if the SCS of the same size is made of elastoplastic material, the signal on the output bar will oscillate significantly during the impact loading process. Since PP is a viscoelastic-plastic material, the stress oscillation during the loading process is due to the back-and-forth propagation of the stress wave in the SCS. The wave system diagram is shown in Figure 9. Such a stress oscillation phenomenon results from the structural characteristics of the SCS, thereby causing no effect on the accuracy and reliability of the compression-shear method in obtaining the final yield point of the material.

## 3. Discussion

### 3.1. Effect of Pulse Width

According to the stress wave theory, in SHPB experiments, bullet length is an important parameter that determines the time of the pulse loaded onto the specimen. Figure 10 shows the comparison between the results of the two experiments on the 45° SCS. In the two experiments, the parameters are as below: bullet length 300 mm, impact velocity 5.47 m/s; bullet length 600 mm, impact velocity 5.64 m/s.

As can be seen from Figure 10, the incident strain signals in the incident bar had roughly the same intensity in the two experiments. The signal response of the semiconductor strain gauge on the output bar was that the intensity of the signals in the transmission bar dramatically decreased at the loading time of 120 us for a bullet with a length of 300 mm. This suggests that the arrival of the unloading wave at the position of the strain gauge was responsible for the decrease. For a bullet with a length of 600 mm at the same velocity, the intensity of the strain signals in the transmission bar underwent a dramatic decrease at the loading time of approximately 141 us. Since the pulse width of the stress loaded was 240 us, the possibility that the strain signal decrease in the transmission bar was caused by the unloading wave can be excluded, and therefore, it should be attributed to the occurrence of the yield or basic instability. In fact, in the dynamic compression experiment, the SCS had been found to undergo sharply decreased stress caused by deformation-induced local instability after it began to yield. Therefore, the reason for the stress drop in the dynamic impact experiment should be the deformation-induced local instability of the specimen.

The analysis of the experimental waveforms for the two bullet lengths revealed that investigating the yield characteristics of the SCS requires the use of a long bullet to perform impact experiments. Should a short bullet be used, the specimen might begin to unload before yielding. As such, the use of a short bullet with a length of 300 mm for the impact test will lead to the failure in achieving the experimental goal of measuring the yield of the SCS.

### 3.2. Influence of Strain Rate Effect

According to the one-dimensional stress wave theory, the bullet velocity determines the specimen deformation rate in SHPB experiments. For SCSs with the same inclination angle, different impact velocities mean different loading rates, which, in turn, result in varying strain rates. The comparison between the impact experiment results of SCSs with different inclination angles showed that similar yield and instability occurred in the SCSs and that with the increasing impact velocity, i.e., the growing strain rate, the deformation of the SCSs underwent a three-stage transition, namely “unyield→yield→failure”. In view of this, the experimental results of the 45° SCS here are taken as the example to investigate the influence of strain rate effect on the deformation behavior and yield law of SCSs and discuss their load force and deformation law under different impact loading rates.

Figure 11 shows the three-stage deformation trends of the 45° SCS under different impact velocities, from which the effect of velocity variations on the load force–time curve can be clearly seen. When the impact velocity was 1.70 m/s, the specimen did not go into yield and therefore, underwent no force drop. However, with the increase of the velocity, the load force peaked at about 150 us and then decreased, indicating that the SCS began to yield. As the velocity further increased (exceeding 5 m/s for the 45° SCS), the peak load force appeared earlier, followed by a larger drop, indicating that the SCS had developed from yield to failure.

As can be seen from the experimental load–displacement curves of the 45° SCS at different impact velocities in Figure 12, before yield and instability occurred, the load force–displacement curve of the unyielding specimen was approximately a viscoelastic response, whose slope decreased with deformation development but increased with the impact velocity. The slop here means the nominal modulus, which is considered to represent the relationship between equivalent stress and equivalent strain considering the homogeneity of the material used to produce the SCS that leads to the load force and displacement necessary to be further decomposed into stress and strain. The elastic modulus of PP still had a certain strain rate effect in the viscoelastic stage, i.e., the modulus changed positively with the loading velocity. After the specimen reached the yielding stage, the displacement corresponding to the peak load became larger, and then the load force decreased significantly with deformation development, which was ascribed to the occurrence of the yield. As the impact velocity was further increased, the specimen transitioned from yield to failure. For example, for the 45° SCS, the experimental curves at impact velocities 5.06 m/s and 5.64 m/s were greatly different from other experimental curves; and the displacement corresponding to the peak load was larger, and the load appeared to drop sharply as the deformation developed.

To further expound the development of the SCS from yield to failure, two experiments with different impact velocities for the 45° SCS were analyzed. Figure 13 presents the load–time curve of 45° SCS failure (v = 5.64 m/s) and yield without failure (v = 3.46 m/s) are given, respectively. By comparison, it can be seen that the curve at v = 5.64 m/s had the maximum signal oscillation amplitude at approximately 25 us during the rising process. This is probably attributed to the local failure and damage the specimen had undergone during the deformation process, which caused the reduced carrying load. When the specimen began to yield, the load dropped rapidly at an amplitude *dF2* = 1390 N, much larger than *dF1* = 325 N, the drop amplitude when v = 3.46 m/s. Meanwhile, the retrieved specimen in the v = 5.64 m/s experiment was found to have completely fractured. The results of the v = 5.64 m/s experiment, as shown in Figure 12, also showed that the load–displacement curve dropped rapidly after reaching the peak value, with the retrieved specimen also completely fractured. All of these indicate that an increased impact velocity will induce the SCS to develop from yield to failure, and that damage-induced failure is the reason for the remarkable drop in the load–displacement curve.

### 3.3. Effect of Stress Path on Yield Surface Development

In order to identify the effect of the stress path on the PP dynamic impact experiment, it is necessary to compare the experimental results of the SCSs with different inclination angles and to adopt the same bullet impact velocity in the experiments on different specimens.

The data distribution of the PP SCSs on the stress plane under dynamic impact loading was obtained by organizing the experimental data, as shown in Figure 14. A loading path that has a constant ratio of the normal compressive stress component to the shear stress component on the plane was obtained by the experiment on SCSs with a fixed inclination angle. By changing the inclination angle, different loading paths were achieved in a similar way. Figure 14 presents the loading paths determined by the angles.

Determining the yield surface requires points on different compression-shear loading paths. Due to the limitation of the experimental conditions, the impact velocities in dynamic experiments are relatively discrete. However, the determination of the yield surface cannot be achieved without a set of experiments under basically the same impact velocity. This is because the impact velocity primarily determines the loading rate if the experimental conditions remain unchanged. Given that the quasi-static experiment revealed the dependence of the yield surface position on the loading rate, the position of the yield surface was analyzed using the experimental data under an impact velocity of about 3.7 m/s, with the intention of avoiding yield surface parameter errors caused by loading rate difference in the dynamic experiments.

The Hu–Pae yield criterion was utilized to obtain the dynamic Hu–Pae yield surface of PP through a fitting, namely the “asterisk” numerical points and the dashed line given by point fitting in Figure 14. By comparing the force–displacement curves, the yield position is found to increase with the inclination angle, which is consistent with the results of the quasi-static experiment [23]. In addition, the corresponding dynamic Von-Mises yield surface is provided in the figure for comparison. To describe the evolution of the yield surface, the static Hu–Pae yield surface and the static Von-Mises yield surface given by quasi-static experiments are also presented in Figure 14. As illustrated, the yield surface developed outward as the loading rate increased. The experimental results under different loading strain rates were fitted, revealing that the Hu–Pae yield criterion only changed the value of the parameter *a_0_*, and that the predicted yield surface was roughly consistent with the experimental results. In the research on the yield surface of Nylon 101, Farrokh et al. [11] identified the relationship between the parameter *a_0_* of the Hu–Pae yield criterion and the strain rate, i.e., $a_{0}=a_{0}^{*}{\left(\frac{\dot{\varepsilon }}{\dot{\varepsilon }*}\right)}^{\beta *}$. Based on the static and dynamic results of the SCS, the parameters of PP were worked out: $\dot{\varepsilon }$* = 1 s^−1^, β* = 0.054,$\;a_{0}^{*}$ = 27.11 MPa, $a_{1}$= 0.42 MPa^−1^, $a_{2}$ = 0.42 MPa^−1^. Hence, the Hu–Pae yield criterion can be written as:$$\sqrt{J_{2D}}=27.11{\left(\dot{\varepsilon }\right)}^{0.054}+\overset{2}{\underset{i=1}{\sum }}\alpha_{i}J_{1}^{i}$$
where $J_{1}$ is the first stress invariant, $J_{1}=\sigma_{xx}+\sigma_{yy}+\sigma_{zz}$, $J_{2D}$ is the second invariant of the stress deflection tensor, $J_{2D}=\frac{1}{6}\left[{\left(\sigma_{xx}-\sigma_{yy}\right)}^{2}+{\left(\sigma_{yy}-\sigma_{zz}\right)}^{2}+{\left(\sigma_{xx}-\sigma_{zz}\right)}^{2}\right]+\left(\tau_{xy}^{2}+\tau_{xz}^{2}+\tau_{yz}^{2}\right)$. 

## 4. Conclusions

In order to investigate the yield behavior of PP under dynamic combined compression-shear loading, five SCSs with different inclination angles were designed and produced, and impact loading experiments with different loading rates were carried out on the specimens. The load–displacement, normal stress–strain, and shear stress–strain curves of the specimens were obtained by processing the strain signals in the dynamic impact experiments. The experimental results showed that:(1)In terms of mechanical response, PP under combined compression-shear loading was of visco-elastic plasticity. The deformation of PP underwent a three-stage transition, namely “unyield→yield→failure”, in which the yield was sensitive to the strain rate.(2)The use of the Hu–Pae yield criterion could more accurately and reasonably describe the yield behavior of PP. The model parameters of the Hu–Pae yield criterion were obtained by fitting the SCS experimental data so that the relationship between PP yield surface development and the strain rate was ascertained.(3)The combined compression-shear loading experiment on the SCS contributes to deepening the research on the mechanical response characteristics of PP-based materials, ascertaining that the yield characteristics of PP are not only affected by the hydrostatic pressure, but also by the stress path.

## Figures and Tables

**Figure 1 polymers-14-03225-f001:**
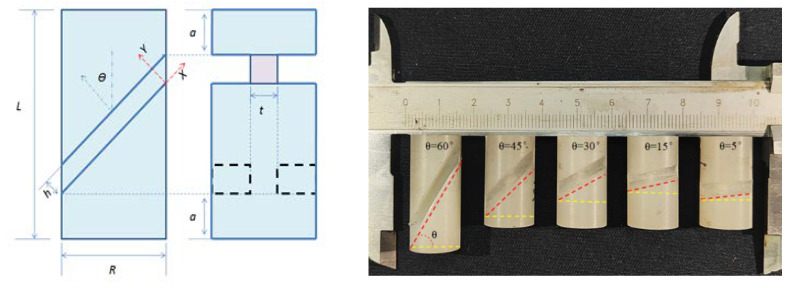
The SCSs of PP.

**Figure 2 polymers-14-03225-f002:**
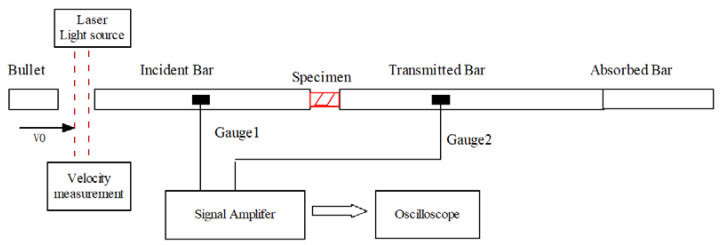
Experimental configuration diagram.

**Figure 3 polymers-14-03225-f003:**
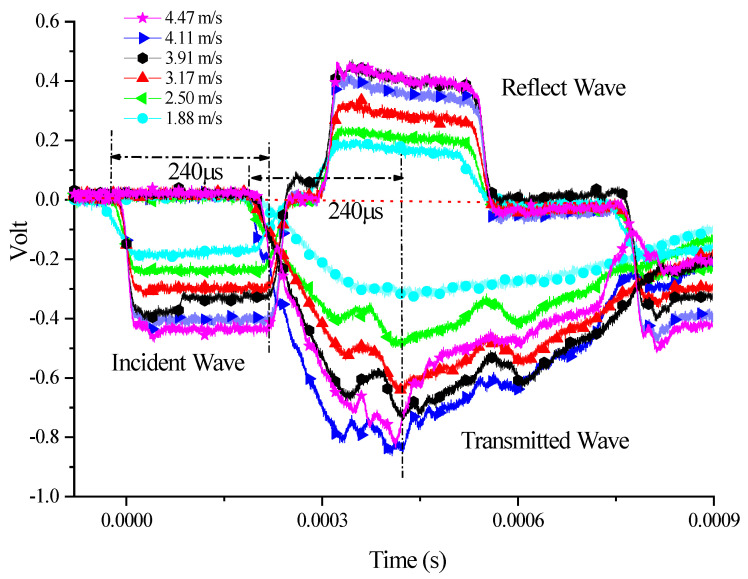
Experimental waveform of 15° SCSs at different impact speeds.

**Figure 4 polymers-14-03225-f004:**
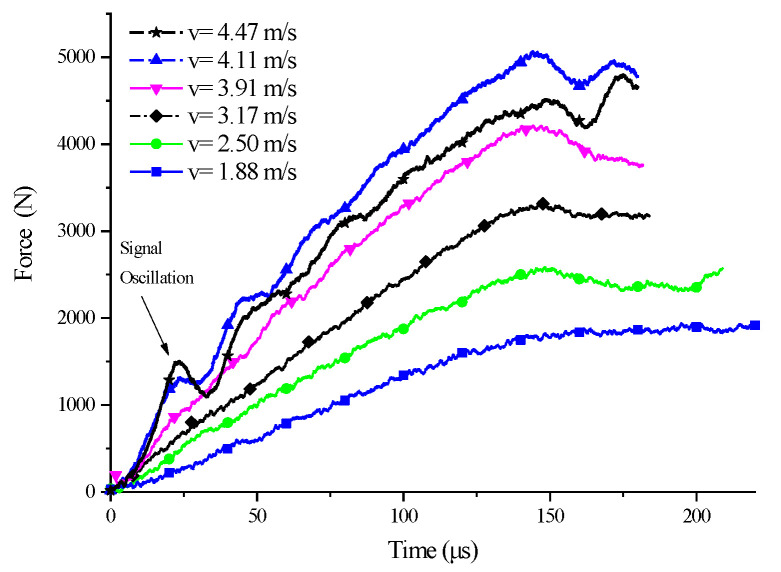
Force–time curve of 15° SCSs at different impact speeds.

**Figure 5 polymers-14-03225-f005:**
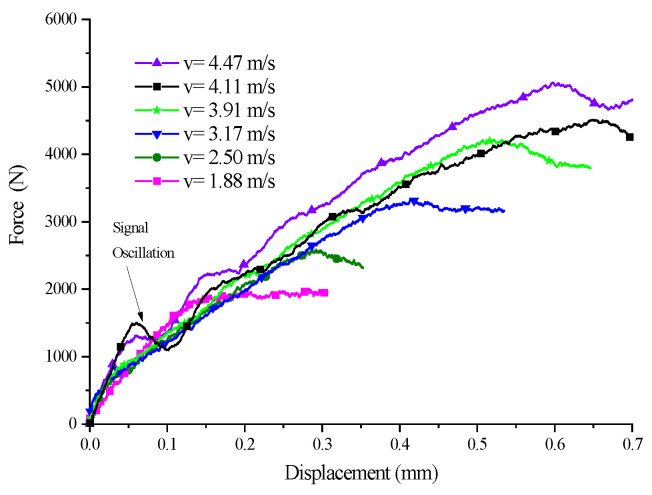
Force–displacement curve of 15° SCSs at different impact speeds.

**Figure 6 polymers-14-03225-f006:**
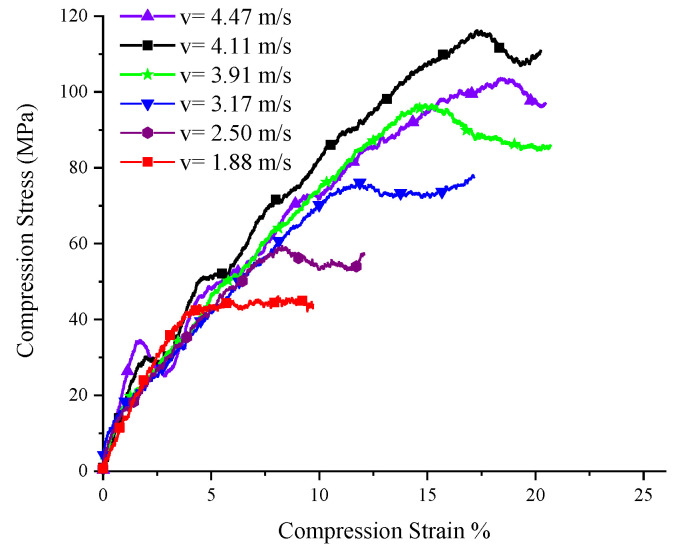
Normal stress–strain of 15° SCSs at different impact speeds.

**Figure 7 polymers-14-03225-f007:**
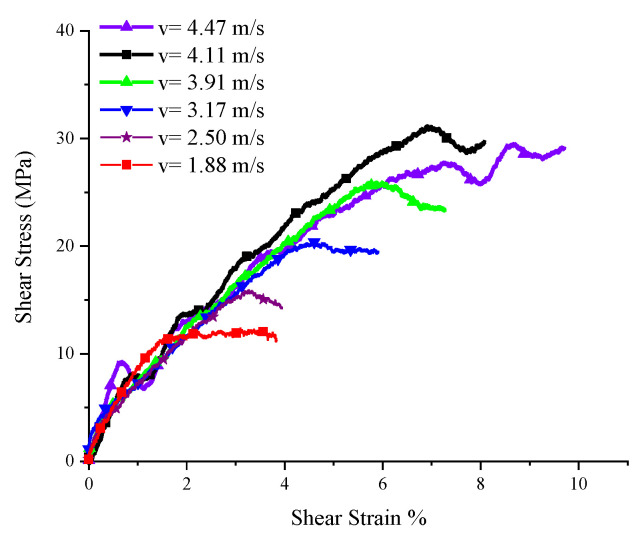
Shear stress–strain of 15° SCSs at different impact speeds.

**Figure 8 polymers-14-03225-f008:**
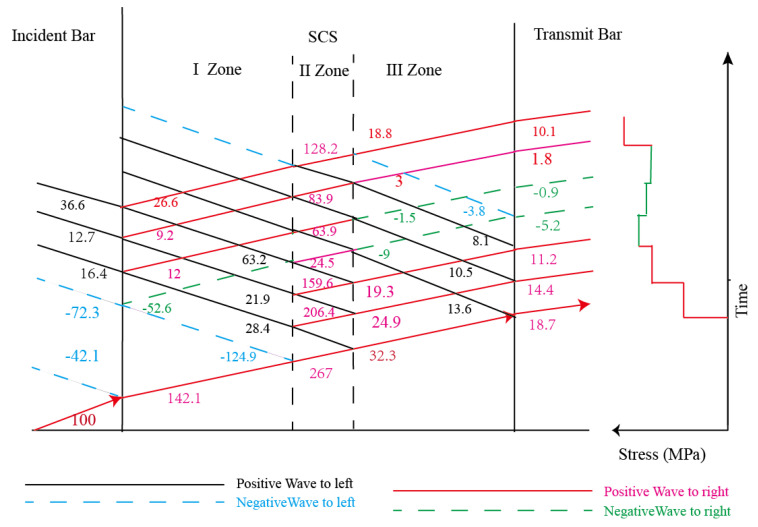
Stress wave propagation based on stress wave theory.

**Figure 9 polymers-14-03225-f009:**
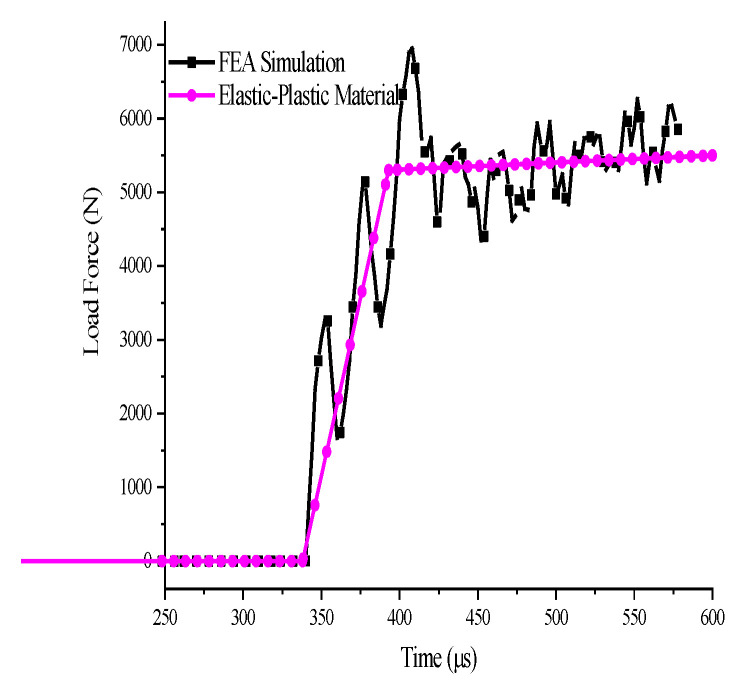
FEA result of PP SCS: load–time curve.

**Figure 10 polymers-14-03225-f010:**
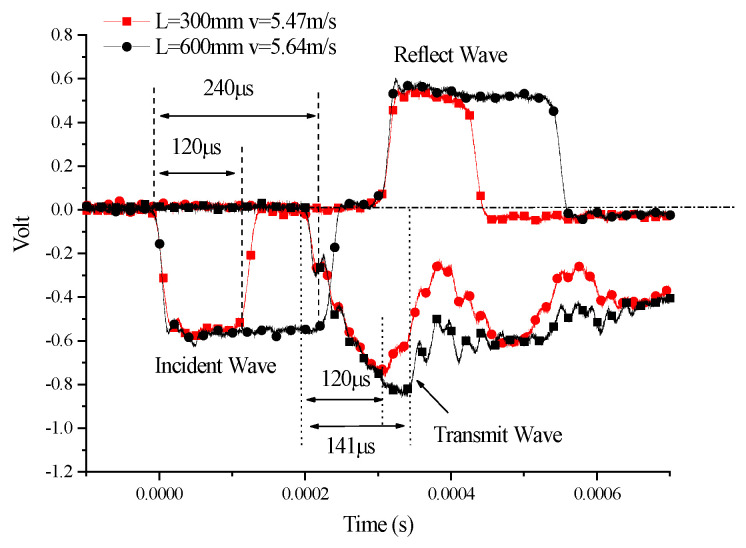
Effect of bullet length of 45° SCSs.

**Figure 11 polymers-14-03225-f011:**
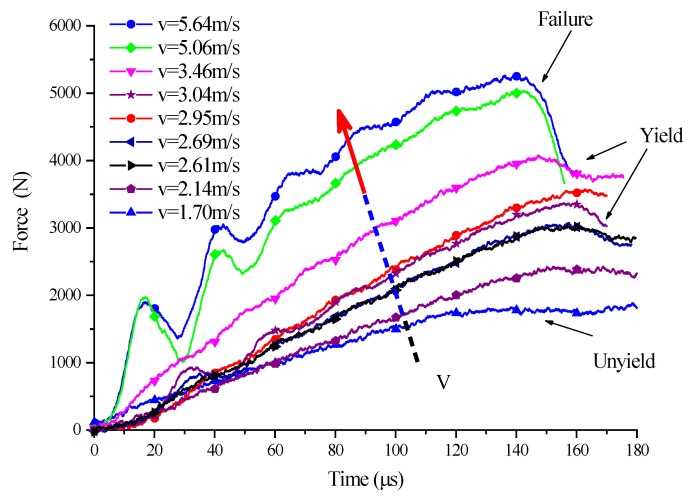
Load–time curves of 45° SCS at different impact speeds.

**Figure 12 polymers-14-03225-f012:**
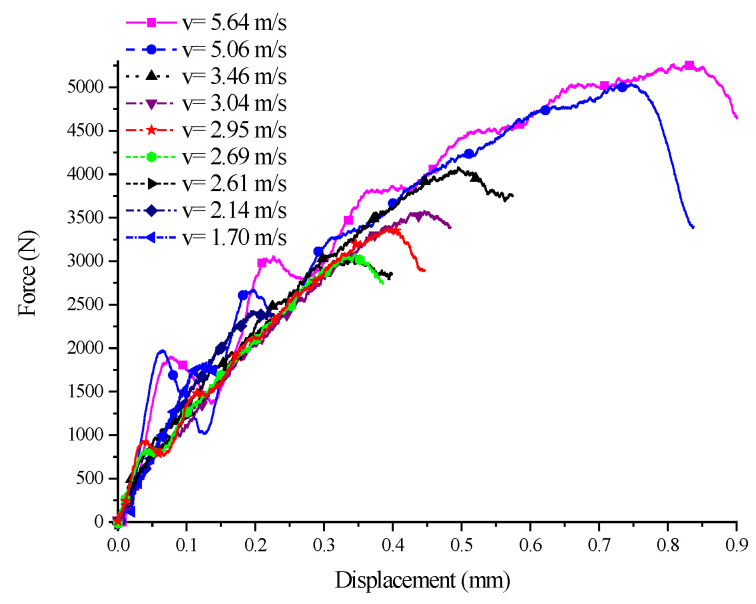
Load–displacement curves of 45° SCS at different impact speeds.

**Figure 13 polymers-14-03225-f013:**
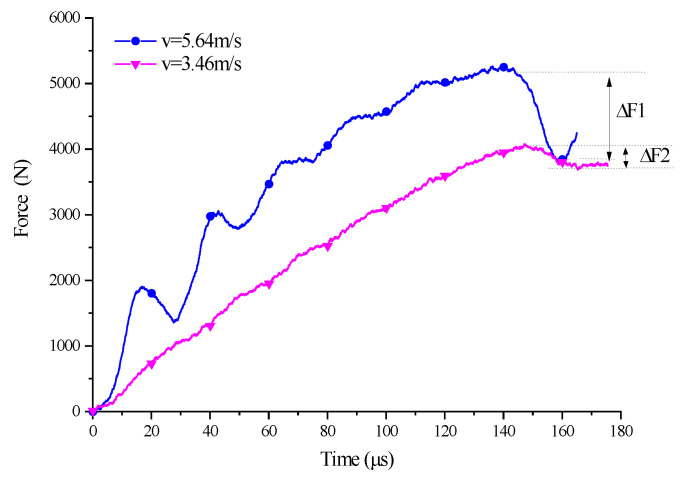
Load–time curve of 45° SCS at v = 4.62 m/s and v = 3.78 m/s.

**Figure 14 polymers-14-03225-f014:**
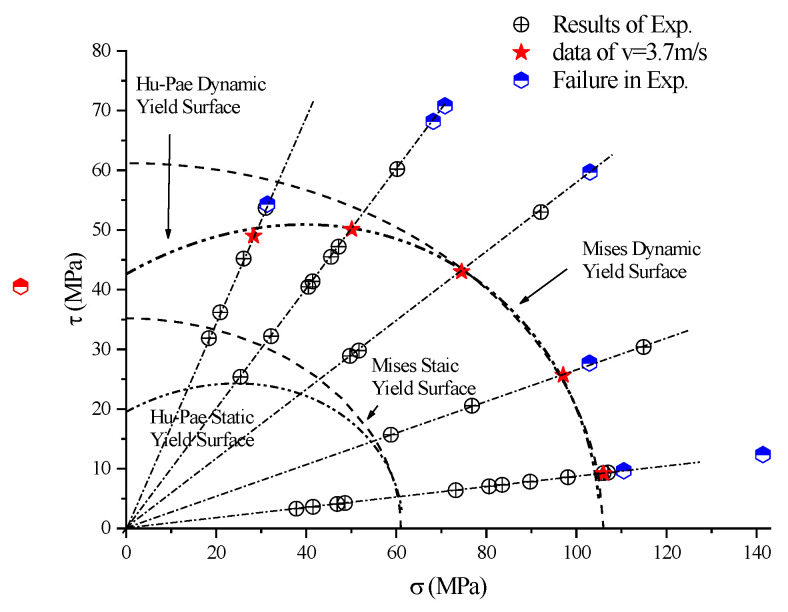
Loading path and evolution of yield surface.

**Table 1 polymers-14-03225-t001:** Material parameters of polypropylene (from HuaSheng Rubber Co. Ltd.).

Material	Density	Tg	Tm	Crystallinity Ratio
Polypropylene	0.907 g/cm^3^	263 K	438 K	64.6%

**Table 2 polymers-14-03225-t002:** Geometric parameters of SCS.

Inclination θ°	Length (mm)	h (mm)	a = (L-h*canθ-R*tanθ)/2 (mm)
5	25	2.01	10.88
15	25	2.07	9.59
30	25	2.31	7.30
45	25	2.83	4.09
60	32	4.0	1.88

**Table 3 polymers-14-03225-t003:** Experimental conditions of SHPB Loading.

	Diameter	Length	Material	Gauge
Bullet	20 mm	600 mm	Ly12	
Incident bar	20 mm	1000 mm	Ly12	Constantan strain gauge
Transmitted bar	20 mm	1000 mm	Ly12	Semiconductor strain gauge

**Table 4 polymers-14-03225-t004:** Results of SCS under dynamic loading.

Angle	Velocity m/s	Max Load N	Max Disp mm	$\sigma_{yy}$ **MPa**	$\sigma_{xy}$ **MPa**	Remarks
5	1.70	1666	0.005			
	1.74	1734	0.006	41.5	3.63	
	1.84	1979	0.019	46.9	4.1	
	1.85	2020	0.02	48.6	4.25	
	2.64	3065	0.02	73.2	6.4	
	2.80	3386	0.03	80.6	7.05	
	2.96	3487	0.03	83.5	7.31	
	3.12	3736	0.03	89.7	7.84	
	3.22	4111	0.03	98.1	8.58	
	3.70	4433	0.07	106	9.27	
	3.97	4458	0.07	107	9.36	
	4.51	4632	0.07	110.5	9.67	Failure
	5.92	5926	0.17	141.4	12.37	Failure
15	1.88	1840	0.02	37.16	9.96	
	2.50	2563	0.03	58.8	15.7	
	3.17	3342	0.07	76.8	20.56	
	3.91	4194	0.14	97.07	25.68	
	4.11	5145	0.165	114.9	30.4	
	4.47	5119	0.18	102.9	27.7	Partial failure
30	2.34	2650	0.03	49.7	28.9	
	2.78	2752	0.03	51.7	29.8	
	3.55	4081	0.07	74.5	43	
	4.34	4741	0.097	92.1	53	
	5.47	5477	0.14	103	59.7	Failure
45	1.70	1898	0.02	25.4	25.4	
	2.14	2376	0.03	32.2	32.2	
	2.61	3002	0.04	40.5	40.5	
	2.69	3067	0.04	41.4	41.4	
	2.95	3411	0.04	45.5	45.5	
	3.04	3509	0.04	47.2	47.2	
	3.46	4010	0.04	52.1	52.1	
	4.20	4519	0.07	60.1	60.1	
	5.06	5031	0.09	68.2	68.2	
	5.64	5269	0.11	70.8	70.8	
60	2.45	2524	0.01	18.4	31.9	
	2.65	2824	0.02	20.9	36.2	
	3.64	3755	0.04	28.3	28.3	
	3.78	3480	0.04	26.1	26.1	Partial failure
	4.16	4155	0.07	31.0	31.0	
	4.40	4211	0.09	31.4	31.4	Failure

**Table 5 polymers-14-03225-t005:** Interface parameters of stress wave propagation analysis.

Wave to Right	Incident Bar/SCS I	SCS I/SCS II	SCS II/SCS III	SCS III/Transmit Bar
Transmission coefficient	1.42	1.88	0.12	0.58
Reflection coefficient	−0.42	−0.88	0.88	0.42
Wave to left	Incident Bar/SCS I	SCS I/SCS II	SCS II/SCS III	SCS III/Transmit Bar
Transmission coefficient	0.58	0.12	1.88	1.42
Reflection coefficient	0.42	0.88	−0.88	−0.42

**Table 6 polymers-14-03225-t006:** Model parameters of finite element analysis (FEA).

	Length mm	Diameter mm	Finite Element Type	Element Number	Material	Modulus GPa	Yield Stress MPa
Bullet	600	20	C3D8R	36,600	LY12	70	400
Incident bar	1000	20	C3D8R	466,000	LY12	70	400
SCS	25	20	C3D4	37,783	Polypropylene	3.6	90
Transmitted bar	1000	14	C3D8R	61,000	LY12	70	400

## Data Availability

Not applicable.

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
