# Peer review of "Experimental Study of Yield Surface in Polypropylene Considering Rate Effect: Stress Path Dependence"

_polymers, 2022, doi:10.3390/polym14153225_

Round 1

Reviewer 1 Report

This is a timely effort of authors to conduct this kind of research on yield surface study for PP by considering the rate effect. However, few points must be addressed before getting this manuscript accepted.

1. Novelty is ambiguous especially due to the presence of similar research papers presented in the literature or introduction sectiion.

2. Were the samples standardised as per some rigid standard of BS or ASME.

3. Dynamic word for testing is inappropriate and should be replaced with quasi static testing.

4. The testing procedures involving bullets with different speeds is seemingly non standardized.

5. If possible then provide the real schematic of bullet test procedure to get more clarity of the testing procedure.

Author Response

We appreciate the reviewers for their careful reading of our manuscript and useful suggestions on how to improve our presentation. The following are our answers and explanations of the issues raised by the reviewers. The revised portions are highlighted in red in the revised manuscript.

1. Novelty is ambiguous especially due to the presence of similar research papers presented in the literature or introduction sectiion.

Answer: Thank you for your attention and suggestion, we have modified the introduction sectiion. The applicability of yield criteria and the evolution law of yield surfaces of polymers in a complex stress state remain unclear. The evolution law of PP’s yield surface under dynamic impact and its relationship with the strain rate havn't get attention. This study achieves dynamic combined stress loading paths under different ratios of compression-shear stress using the SHPB. And the dynamic yield surface of the specimens is obtained according to the hydrostatic pressure-dependent Hu-Pae yield criterion, and their rate effect is analyzed as well.

2. Were the samples standardised as per some rigid standard of BS or ASME.

Answer: Thank you for your proposal and patience and also thank you for making our paper readable and valuable. The samples were standardised as the rigid plastics of ASTM D695.

3. Dynamic word for testing is inappropriate and should be replaced with quasi static testing.

Answer: As suggested, we have modified the related words.

4. The testing procedures involving bullets with different speeds is seemingly non standardized.

Answer: Thank you for your suggestion. The statements about the testing procedures involving bullets with different speeds have been added. When the bullet passes through the laser light path and block the laser light in turn, so that the photoelectric sensor can obtain the signal. The speed V0 can be measured by using two photoelectric sensors at different positions. After the bullet hits the incident bar, the stress wave propagates to the SCS specimen, transmission bar and absorption bar in turn, and the reflection occurs at the interface. 

5. If possible then provide the real schematic of bullet test procedure to get more clarity of the testing procedure.

Answer: As suggested, Figure 2 has been replaced.

Reviewer 2 Report

The referee would like to recommend this work to major revision and to be published after consideration according to the comments below:

1.      Please add more current papers in the literature and improve introduction section. Some interesting papers related to the topic of this manuscript could be:

Experimental and numerical study on HDPE/SWCNTnanocomposite elastic properties considering the processingtechniques effect. Microsystem Technologies, (2020), 26, 24232441.

Experimental studies on elastic properties of high density polyethylene-multi walled carbon nanotube nanocomposites. Steel and Composite Structures, (2021), 38(2), 177-187.

Author Response

We appreciate the reviewers for their careful reading of our manuscript and useful suggestions on how to improve our presentation and make our paper readable and valuable.

1.      Please add more current papers in the literature and improve introduction section. Some interesting papers related to the topic of this manuscript could be:

Experimental and numerical study on HDPE/SWCNTnanocomposite elastic properties considering the processingtechniques effect. Microsystem Technologies, (2020), 26, 2423–2441.

Experimental studies on elastic properties of high density polyethylene-multi walled carbon nanotube nanocomposites. Steel and Composite Structures, (2021), 38(2), 177-187.

Answer: As suggested, we added these references to the sentence in the introduction of the revised manuscript.